# Friend or Foe? The Endophytic Fungus *Alternaria tenuissima* Might Be a Major Latent Pathogen Involved in Ginkgo Leaf Blight

Xiaojia Su [1,2], Ruirui Shi [1,2], Xiaobo Li [1], Zine Yu [1,2], Linfeng Hu [3], Haiyan Hu [1,2], Meng Zhang [1], Jingling Chang [1] and Chengwei Li [1,2,4,*]

1   College of Life Science and Technology, Henan Institute of Science and Technology, Xinxiang 453003, China; mengz1115@hist.edu.cn (M.Z.); changjl001@126.com (J.C.)
2   Henan Engineering Research Center of Crop Genome Editing, Henan International Joint Laboratory of Plant Genetic Improvement and Soil Remediation, Xinxiang 453003, China
3   College of Chemistry and Chemical Engineering, Henan Institute of Science and Technology, Xinxiang 453000, China
4   College of Biological Engineering, Henan University of Technology, Zhengzhou 450001, China
*   Correspondence: lcw@haut.edu.cn

**Abstract:** Ginkgo leaf blight, one of the most economically important ginkgo diseases, has become very prevalent in many places in China. Flavonoids and endophytes are both considered important in ginkgo plant functioning. However, little is known about the potential relationships among ginkgo leaf blight pathogens, flavonoid accumulation profiles in infected leaves, and ginkgo leaf endophytes. In this study, the flavonoid accumulation profiles in infected leaves, pathogens of ginkgo leaf blight, and the endophytes of healthy ginkgo leaves were characterized. The levels of total flavonoids in the healthy parts of the infected leaves were significantly higher than those in the healthy leaves. Furthermore, *Alternaria tenuissima*, *Botryosphaeria dothidea*, and *Dothiorella gregaria* were identified as pathogens of ginkgo leaf blight; among them, *A. tenuissima* was the major pathogen. The in vitro experiments showed that flavonoids (apigenin, luteolin, and kaempferol) could significantly inhibit the growth of one or more pathogens at a concentration of 10 mg/L. Furthermore, fifty-six ginkgo leaf endophytic fungi (GLEF) from healthy ginkgo plants were isolated and characterized. Among them, *Alternaria* spp. were the most abundant, and GLEF55 shared the same ITS sequence with the pathogen *Alternaria tenuissima*. Thereafter, four flavonoid-producing endophytes were selected and their effects on the growth of pathogens were evaluated. The extracts of GLEF55 could significantly inhibit the growth of the pathogens *B. dothidea* and *D. gregaria* simultaneously in vitro, but not the growth of the pathogen *A. tenuissima*. Furthermore, the dual cultures of the candidate GLEF and ginkgo leaf blight pathogens revealed that GLEF55 had a similar growth rate to that of *A. tenuissima* and *D. gregaria*, but its growth rate was significantly slower than that of *B. dothidea*. Finally, the GLEF exhibited variable roles when facing pathogens in ginkgo leaves. Among them, GLEF55 showed similar pathogenicity as the pathogen *A. tenuissima* when they were dually cultured in ginkgo leaves. By contrast, GLEF17 (an uncultured soil fungus) could significantly counteract the pathogenic effects of *A. tenuissima* and *D. gregaria*, but it dramatically exacerbated the pathogenic effects of *B. dothidea*. Larger lesion areas were observed on the side of ginkgo leaves where GLEF39 (*Alternaria* sp.) or GLEF54 (*Aspergillus ruber*) and pathogens were simultaneously inoculated, which suggested that the pathogenicity of specific endophytic fungi occurred when plants were wounded. Overall, *A. tenuissima*, a major pathogen of ginkgo leaf blight, might lurk inside the plants as a friendly endogenous fungus and convert into a hostilely pathogenic mode at a particular time. This study proposed a possible cause of ginkgo leaf blight and provided potential theoretical guidance for its prevention.

**Keywords:** dual culture assay; endophytic fungi; flavonoids; ginkgo leaf blight; pathogens

## 1. Introduction

*Ginkgo biloba* (L.) is one of the most ancient plants and holds important values in horticulture and clinical medicine. Ginkgo leaf blight is regarded as one of the most destructive foliar diseases of *G. biloba* and is typically characterized by leaves being abnormally scorched and wilted, shedding of leaves, and even the death of the infected tree. This disease results in serious damage to the economic value of the ginkgo industry [1]. Moreover, along with the extensive planting of ginkgo in China, leaf blight has become very prevalent and presents serious trends in many places of China, such as Beijing, Jiangsu, Sichuan, Guangxi, and Northeastern China [1–8].

Ginkgo leaf blight begins with abnormal wilting at the leaf margin between May to June, and then the infected leaves gradually wither and shed until August and September [8,9]. Gaining insight into the etiology of this disease is critical to preventing it from happening. There are two different opinions about the etiology of this disease. The prevalent opinion is that a leaf pathogen is the main cause of the disease. This opinion is supported by investigations that have characterized several pathogens from ginkgo leaf blight-infected leaves. Zhu and Shi [10] studied ginkgo leaf blight pathogens in Jiangsu Province (China) from 1986 to 1989 using pathogenicity tests, and *Alternaria* sp., *Colletotrichum* sp., and *Pestalotia* sp. were identified, among which *Alternaria* sp. was the main pathogen. In addition, You et al. [8] showed that ginkgo leaf blight pathogens varied and influenced throughout the sampling time except *Alternaria* sp. *Phyllosticta* sp. and *Fusarium* sp. were also isolated from infected leaves in Nanjing (Jiangsu Province, China) in 2021 [8]. However, validation studies via strict pathogenicity tests of the isolated candidate pathogens were not performed by You et al. [8]. Furthermore, based on the three pathogens identified by Zhu and Shi [10], some fungicides (e.g., Rovral, thiophanate-methyl, carbendazim, and hexaconazole) were proven to be effective in vitro and contributed up to an 85% prevention rate of ginkgo leaf blight in a field experiment [2,7,11].

In contrast, another opinion is that the weakness of ginkgo trees is caused by poor cultivation and management practices, which should be paid more attention to. Fan [5] proved that the effects of fungicides are very limited in alleviating the occurrence of this disease, but proper cultivation management is more effective based on a 4-year disease control practice on ginkgo leaf blight in Shenyang, China. This suggest that a weak ginkgo tree offers opportunity for invasion by ginkgo leaf blight pathogens, but fungicides could not effectively eradicate pathogens. Additionally, *Alternaria* sp. has proven to be the main pathogen throughout the course of ginkgo leaf blight, and is very difficult to control [8,9]. Therefore, He et al [9] speculate that whether *Alternaria* sp. has long existed in the ginkgo leaf bud before spring? This raises other questions: where do these pathogens especially *Alternaria* sp. come from? Does the disease start with an imbalance of endophytes in a weak plant?

Previous research has shown that a symbiotic continuum from symbiosis to parasitism exists between plants and endosymbiotic fungi, which is affected by many factors, including the node of transformation, the mode of infection, plant age, environmental conditions, and genetic background [12,13]. Fungal endophytes may turn pathogenic during host senescence, although they are considered to be uninjurious in healthy plants [14–16]. This suggests that a symbiotic system is not always constant, and it significantly depends on the status of the host.

Endophytes play important roles in plant functioning, some of which are considered to protect their host from infectious diseases by regulating the synthesis and accumulation of secondary metabolites in plants [12,17–22]. Therefore, endophytes have been recognized as important sources of a variety of novel secondary metabolites, including antifungal compounds [23–26]. For example, an endophytic bacterium, *Bacillus subtilis* DZSY21, isolated from the leaves of *Eucommia ulmoides* oliv. showed biocontrol effects on the southern corn leaf blight by inhibiting *B. maydis* by accumulating antifungal lipopeptides and activating an induced systemic response [27]. An endophytic fungus, *Piriformospora indica*, could

enhance the drought tolerance of trifoliate orange by modulating the antioxidant defense system and the composition of fatty acids [28].

Previous research has focused on promising endophytes and ginkgo leaf blight pathogens [8,10,26]. Little attention has been paid to the way that plant metabolites change during the infection of ginkgo leaf blight pathogens. Therefore, it is interesting to explore the possible relationships among host plant metabolites, endophytic fungi, and ginkgo leaf blight pathogens. To achieve this, we characterized the flavonoid profiles of infected ginkgo leaves, ginkgo leaf blight pathogens, and the endophytic fungi of healthy ginkgo leaves. Furthermore, the growth of pathogens was evaluated after exposure to the metabolites from plants and endophytes in vitro. Finally, the roles of candidate endophytes in the occurrence of ginkgo leaf blight on ginkgo leaves were investigated using dual culture assays and co-inoculation.

## 2. Materials and Methods

### 2.1. Plants and Growth Conditions

Mature ginkgo (*G. biloba* L.) seeds with mesosperm were potted in roseite with deionized water in a climate-controlled chamber under a 16 h light/8 h dark photoperiod for germination, and about eighty seeds were used. After germination, sixty-five ginkgo seedlings were obtained and transferred into soil to develop for 1.5 months.

In total, about fifty mature ginkgo seeds (with fully developed embryos at the cotyledon stage and endosperm) were sterilized using 75% ethanol for 20 min. Then, residual ethanol on the endosperms was removed by washing with sterile water 3–4 times. Finally, complete embryos were carefully separated from the endosperms and immediately planted into solid Murashige and Skoog medium (without any phytohormones) on an ultra-clean workbench. Mature embryos were cultured under a 16 h light/8 h dark photoperiod in a tissue culture room at 22 °C under constant light (150 µmol photons m$^{-2}$ s$^{-1}$). About twenty-eight sterile ginkgo seedlings were developed and obtained after continuous cultivation in the MS medium for 1–1.5 months.

### 2.2. Pathogen Isolation

The ginkgo leaves used for pathogen isolation in this study were collected from Xinxiang city (∼35°18′13.71″ N 113°55′15.05″ E ), Henan province, China. Xinxiang has a warm temperate continental monsoon climate, with an average annual precipitation of 380.8 mm, an average annual temperature of 15.6 °C, and an elevation of 79 m above sea level (http://tjj.xinxiang.gov.cn/sitegroup/root/html/402881bc6c8f1d54017139b4233d0b06/ae6cfebaa16d4381901a4fead1312282.html, accessed on 5 June 2023). As gingko trees are widely distributed across the three universities (Henan Normal University, HNU; Xinxiang Medical University, XMU; and Henan Institute of Science and Technology, HIST) of Xinxiang city, a total of 102 samples (51 healthy/51 infected leaves) from lateral branches at the base of the crown were collected from 17 sites (Figure S1A; Table S1) in 7–8 September 2019, according to the distribution of ginkgo trees. The development of ginkgo leaf blight at different infection stages is shown in Figure S1B. In the healthy stage, the canopy is characterized by abundant and mostly mature leaves with a dark green color, and almost no yellow leaves are visible (Figure S1B(a)). The early stage of ginkgo leaf blight is characterized by the beginning of leaf discoloration. Then, some leaves of the lateral branches in the canopy begin to turn yellow and curl from the top, and the percentage of infected leaves is about 25–50% (Figure S1B(b)). In the middle stage, the leaves of the infected tree almost turn light yellow, and the percentage of infected leaves is about 50–80% (Figure S1B(c)). During the serious stage, canopy leaves of the infected tree are sparse, and even the tree is almost bare (Figure S1B(d–f)). At each site, healthy and infected leaves (20–50% of leaf area) of the same ginkgo plant were sampled at the early infection stage. After collection, the leaves from the same tree were immediately sealed together in self-styled PE bags and quickly transported to the laboratory at 4 °C. Finally, pathogen isolations from the infected leaves were performed in 24 h.

A total of 17 infected leaves (one leaf per tree) were used to isolate ginkgo leaf blight pathogens according to the method used by Chen [7] with some modifications. Briefly, the infected leaves were sterilized with 75% ethanol for 2.5 min and rinsed with sterile water 3 times. Then, the tissues at the disease–health junction sites were cut into $0.5 \times 0.5$ cm$^2$ pieces and cultured on potato dextrose agar (PDA) medium ($\Phi$ 5.5 cm) separately, with the leaves being right side up at 22 °C. The healthy leaves were used as the controls. The fungal colonies present at the margins of the cultured infected leaves were selected as candidate pathogens.

### 2.3. Histology

The infection sites of the ginkgo leaf blight-infected leaves were cut into about 1 cm$^2$ pieces and immersed in a decolorization solution (ethanol/acetic acid ratio of 3:1, V/V) until complete decolorization. Then, the mycelia were observed using a microscope (Axio Zoom.V16, ZEISS, Oberkochen, Germany) after being stained by Coomassie brilliant blue solution for 1 min. The fresh mycelia of the candidate pathogens in the Petri plates were stained with lactophenol cotton blue staining solution (Solarbio, Beijing, China) for 5 min and observed using a microscope under 1700 LP/mm resolution after being rinsed with water. The scale bars for the observations of the infected leaves and mycelia were 200 µm and 50 µm, respectively.

### 2.4. Pathogenicity Determination

The healthy ginkgo leaves and the plantlets grown from cultured embryos were wounded by sterile needles. The mycelial plugs ($\Phi$ 5.0 mm) of isolated pathogens were cut from the edge of the fresh colonies and transferred onto the wound while being upside down. From there, the mycelial plug cuttings continuingly grew and reproduced in the wound for 1–2 weeks until observation, as described by Chen [7]. Plastic bags were put over the ginkgo leaves infected with fungi indoor for isolation, and water was sprayed daily for moisturization. Images were captured after clearing out the residual mycelial plugs on the wounds of the ginkgo leaves.

### 2.5. Endophyte Isolation

Research shows that ginkgo leaf blight begins from May, and then the leaves gradually wither and shed until September [8,9]. Therefore, ginkgo trees that are still in good health in mid-October are considered healthy. Endophytes were isolated from the leaves of healthy ginkgo trees at the three sites (HNU, XMU, and HIST) in mid-October of 2018. In total, 54 leaves from lateral branches in the crown of 9 healthy trees (6 leaves per tree) were sampled and transported to the lab at 4 °C. Endophytes were isolated according to the methods described by Kumar et al. [29] within 24 h. Briefly, the leaves were treated with 75% ethanol for 15 min for surface sterilization and rinsed with sterile water 3 times. Then, the sterile leaves were cut into about $1 \times 1$ cm$^2$ pieces and cultured on MS medium at 22 °C in a tissue culture room under 16/8 (light/dark) photoperiod for 6–14 days. The 3rd rinsing water from the sterile leaves was used as a negative control. The hyphal tips of the developing fungal colonies were transferred to the fresh PDA medium for endophytic purification and identification.

### 2.6. Fungal Identification

Total genomic DNA from the mycelia was extracted according to the method proposed by Luo et al. [30]. Briefly, several fresh fungal mycelia were picked up with a sterile toothpick and quickly put into 50 µL of NaOH solution (50 mmol/L). Then, the mixture was fully vortexed and boiled in water for 10 min to lyse the cells and release the fungal DNA. Then, 1/10 of the original volume (5 µL) of Tris-HCl buffer (1 mol/L, pH = 8.0) was added and mixed. Finally, the mixture was centrifuged at 12,000 rpm for 10 min, and 2 µL of the supernatant was used as the DNA template for each PCR amplification. The internal transcribed spacer (ITS) sequences used for the identification of candidate fungi were

obtained through PCR with 2 × Taq Master Mix (Vazyme, Nanjing, China), up-mentioned DNA templates, and the universal primers ITS1 (5'-TCCGTAGGTGAACCTGCGG-3') and ITS4 (5'-TCCTCCGCTTATTGATATGC-3') [31]. PCR was performed using the following thermocycler program: initial denaturation at 94 °C for 5 min, 35 cycles of denaturation at 94 °C for 30 s, annealing at 58 °C for 30 s, and extension at 72 °C for 30 s, with a final extension step at 72 °C for 5 min. The ITS amplification products were checked on agarose gels (0.8% in 1 × TAE buffer). The amplicons were purified using the TIANgel Midi Purification Kit (TIANGEN BIOTECH, Beijing, China). The purified ITS amplification products were sequenced using the Sanger sequencing method by GeneCreate Biological Engineering Co., LTD (Wuhan, China). Fungi were identified when they had over 95% similarity in ITS sequences compared to the data in the National Center for Biotechnology Information (NCBI) database (https://www.ncbi.nlm.nih.gov/, accessed on 20 March 2023).

*2.7. Phylogenetic Analysis*

Maximum likelihood phylogenetic trees based on the ITS sequences of the identified fungi were constructed using MEGA 6.0 and Clustal W (https://www.genome.jp/tools-bin/clustalw, accessed on 20 March 2023) was used for multiple sequence alignments. Phylogenetic analyses using maximum likelihood, Tamura–Nei model, and the nearest-neighbor-interchange (NNI) ML heuristic method under complete deletion gaps and 1000 bootstrap replicates were performed [32]. The GenBank accession numbers of the ITS sequences are listed in Tables S2 and S3.

*2.8. Pathogen Inhibition Experiments and Dual Culture Assays in Petri Plates*

The inhibitory effects of metabolites on ginkgo leaf pathogens (*Alternaria tenuissima*, *Botryosphaeria dothidea*, and *Dothiorella gregaria*) were evaluated using the methods of mycelial growth rate, as previously described [7,33]. Briefly, mycelial plugs (Φ 5 mm) were cut from the edge of the fresh colonies, transferred onto the fresh PDA medium with flavonoids (isorhamnetin, kaempferol, quercetin, apigenin, and luteolin) at different concentrations (1 mg/L and 10 mg/L) or endophytic flavonoid-like metabolites at 1/1000 dilution (measured as total flavonoid concentration of 0.2–0.79 mg/L), and incubated at 25 °C. The resulting colonies were observed and the diameter of each colony was measured daily. Each isolate was presented by three separate replicates.

Dual culture of ginkgo endophytes (GLEF39, GLEF55, GLEF17, and GLEF54) against ginkgo leaf blight pathogens (*Alternaria tenuissima*, *Botryosphaeria dothidea*, and *Dothiorella gregaria*) was conducted using a dual culture technique [34]. Yeast-extract sucrose (YES) medium (150 g/L of sucrose, 20 g/L of yeast extract, 0.5 g/L of $MgSO_4.7H_2O$, 10 mg/L of $ZnSO_4.7H_2O$, 5 mg/L of $CuSO_4.5H_2O$, and 20 g/L of agar for solid medium) was selected for the dual culture as it is conducive for fungi to produce flavonoid-related compounds [35]. The ginkgo endophytes against ginkgo leaf blight pathogens were inoculated on solid YES medium, opposite of each other on the center line (~2.25 cm) of the Petri plates (Φ 5.5 cm). The resulting conditions of each culture were measured at the final steady antagonism states. As no obvious inhibition zones appeared in some antagonism strains, relative extension length (REL) was used to measure the antagonism states of the endophytes and pathogens in the dual culture. Each pair in the dual culture was presented by three separate replicates. The specific calculation method is as follows:

$$\text{REL of Endophyte in the dual culture bioassay} = Y/X \times 100\%$$

$$\text{REL of Pathogen in the dual culture bioassay} = Z/X \times 100\%$$

where X represents the distance between the center points of an endophyte and a pathogen, and it is constant in dual culture measurement; Y represents the distance between the center of the endophytic fungus to its colony edge along the center line; and Z represents the distance between the center of the pathogenic fungus to its colony edge along the center line.

### 2.9. Total Flavonoid Determination and Metabolite Analysis

To analyze the flavonoid accumulation in the healthy and infected leaves, 17 independently paired (healthy–healthy/sick–sick) samples (leaves from one tree were mixed with each other to form one paired sample per tree) were used. And one sample was divided into three replicates for extraction and determination. The ginkgo leaves were ground into powder with liquid nitrogen and then freeze-dried at $-40\,^{\circ}\text{C}$ for further determination.

Metabolites from fifty-six isolated endophytes were prepared from liquid fermentation. Six $21.6\,\text{mm}^2$ fresh mycelial plugs of each endophyte were inoculated into 10 mL of YES liquid medium (without agar) at $25\,^{\circ}\text{C}$ with a 12 h photoperiod and cultured for 8 days. Then, the mixture was ultrasonicated and extracted for metabolites by adding 10 mL of ethyl acetate. Finally, the upper organic phase was obtained via centrifugation and dried using a nitrogen blower. The dried metabolites were redissolved in 500 μL of dimethyl sulfoxide (DMSO), which was used for total flavonoid determination and metabolite analysis. Forty out of the fifty-six isolated endophytes showed obvious growth in the YES liquid medium and were used for flavonoid determination. Independent experiments were performed twice, and three replicates were used for each determination. Total flavonoids were quantitatively determined using chemical colorimetric methods according to Su et al. [36]. Briefly, 400 μL of deionized water was added and mixed with 100 μL of the target DMSO mixture. Then, 30 μL of $NaNO_2$ (5%) was added into the mixture. Five minutes later, another 30 μL of $AlCl_3$ solution (10%) was added and mixed. Another 10 min later, 200 μL of 1 M NaOH was added, and the color of the mixture presented white to dark brown according to the content of flavonoids. The reaction mixture was finally filled to 1 mL with deionized water. And the absorbances of the reaction mixture at 510 nm were obtained. Quercetin was used to draw the standard curve. Profiles of flavonol glycosides, flavonol aglycones, and biflavones were analyzed using HPLC-MS/MS as described by Su et al. [37].

### 2.10. Statistical Analysis

Flavonoid accumulation profiles in the ginkgo leaf blight-infected leaves were analyzed using one-way ANOVA. The leaf categories were fixed as the independent factors, and flavonoid contents were set as the dependent variables. Prior to the analysis, the dependent variables were tested using one-sample K–S nonparametric test to test for normality. Furthermore, Levene's test was used to check for the homogeneity of variance. After confirming the data's normality and homogeneity of variance, LSD (L), and Duncan (D) tests were applied to detect significant effects at the alpha level of 0.05.

The differences in the pathogenicity of the candidate pathogens on in vitro ginkgo leaves and ginkgo plants with multi-site inoculations were both analyzed using one-way ANOVA. The types of pathogens were fixed as independent factors, and the relative lesion areas in the ginkgo leaves were set as the dependent variables. Then, Dunnett's T3 test with Brown–Forsythe and Welch values was used to analyze the differences ($p < 0.05$) among the dependent variables because of variance heterogeneity.

Symmetrical inoculations of the candidate pathogens in ginkgo plants and dual culture assays were analyzed using the independent samples *t*-test. Different treatments were fixed as the grouping variables. The values of the relative lesion areas or relative extension length (REL) were set as the tested variables. One-sample K–S and Levene's tests were used to check for normality and homogeneity of variance, respectively. Two-tailed tests of significant differences between groups were analyzed ($p < 0.05$ or 0.01).

Effects of flavonoids and endophytic metabolites on the growth of ginkgo leaf blight pathogens in vitro were analyzed using one-way ANOVA. Treatments with different compounds were fixed as the independent factors. The relative diameters of the pathogen colonies in each group were set as the dependent variables. Furthermore, significant effects were detected at the 0.05 level, and LSD (L) and Duncan (D) tests were applied with the dependent variables. Prior to the analyses, one-sample K–S and Levene's tests were used to check for the normality and homogeneity of variance, respectively. Furthermore, the power (2, x)-transformed data of *Alternaria tenuissima* and *Dothiorella gregaria* treated with endophytic

metabolites were used to meet the normality and homoscedasticity assumptions. All statistical analyses were performed using the SPSS Statistics 17.0 software (IBM, Armonk, NY, USA).

## 3. Results

### 3.1. Flavonoid Accumulation Profiles in Infected Leaves

The leaves infected with ginkgo leaf blight disease were characterized by being yellow, curled, and shriveled from the tip, and the necrosis gradually extended into the interior (Figure 1A). The total flavonoid accumulation profiles in the infected leaves were characterized using chemical colorimetric methods and HPLC-MS/MS. The results showed that the levels of total flavonoids in the healthy–sick (H-S) parts of the infected leaves significantly increased by 18.7% compared to that of the healthy parts (H). Furthermore, the total flavonoids in the sick parts (S) of the infected leaves dramatically decreased by 51.8% compared to the healthy ones (Figure 1B). Further analysis indicated that no significant differences were detected in the categories of total flavonoids (Figure S2); however, the accumulation of specific flavonols (quercetin, kaempferol, and isorhamnetin) significantly increased in the H-S leaves (Figure 1C,D). By contrast, the levels of biflavones (bilobelin, ginkgetin, and isoginkgetin) showed a significant reduction in the infected leaves, except for sciadopitysin (Figure 1C,D).

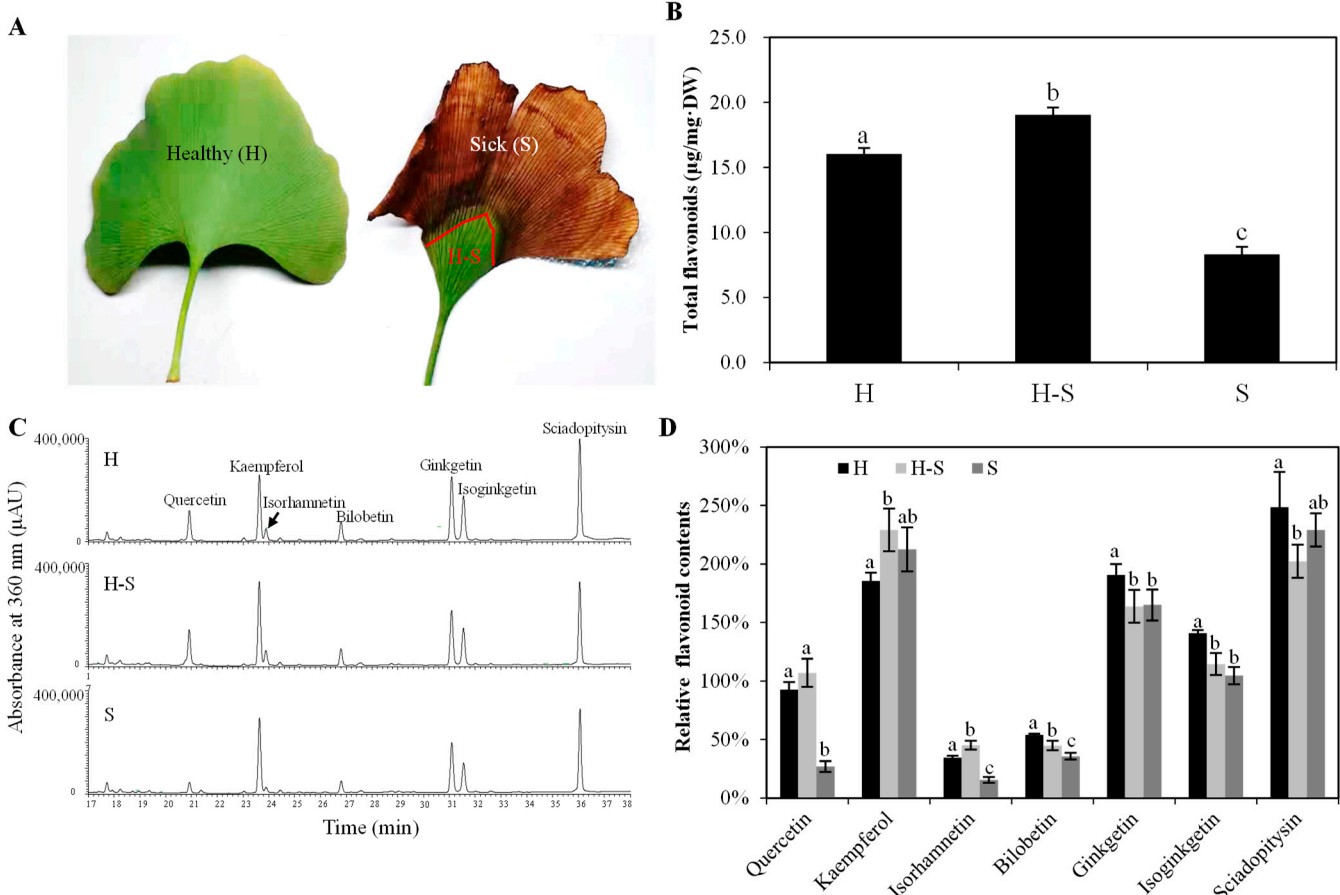

**Figure 1.** Flavonoid accumulation profiles in ginkgo leaves upon leaf blight infection. (**A**) Representative photographs of healthy (**left**) and ginkgo leaf blight-infected leaves (**right**). (**B**) Total flavonoid content in healthy (H), healthy–sick (H-S), and sick (S) leaves based on chemical (5% NaNO$_2$, 10% AlCl$_3$, and 1 M NaOH) colorimetric methods. (**C**) HPLC chromatograms of flavonoid profiles in healthy (upper panel), healthy–sick (middle panel), and sick (lower panel) leaves. Total flavonol aglycones were analyzed via hydrolysis with an equal original volume of 6 N HCl at 70 °C for 40 min,

and 100% methanol at twice the original sample volume was added to stop the reaction. Then, the mixture was injected into HPLC-MS/MS for analysis. (**D**) Quantification of flavonoid contents of (**C**). The area of quercetin in healthy leaves was set to 1. Data are expressed as mean values with standard deviation (S.D.), *n* = 3. Statistically significant differences were tested via one-way ANOVA using SPSS Statistics 17.0. Different letters represent significant differences with an alpha value of 0.05.

### 3.2. Isolation and Identification of Ginkgo Leaf Blight Pathogens

To identify the ginkgo leaf blight pathogens, infected leaves from Xinxiang (Henan province, China) were collected. In total, eight different fungi were isolated from the disease–health junction sites of 17 infected leaves (one leaf per tree) at the early infection stage (Figure S1B(b)). These fungi were used for pathogenic tests with in vitro sterile leaves according to Koch's postulates (Table S4). The leaves infected with *Alternaria tenuissima* (At), *Botryosphaeria dothidea* (Bd), and *Dothiorella gregaria* (Dg) presented typical black or brown wilted patches at the infection sites in vitro (Figure 2A(a–d)).

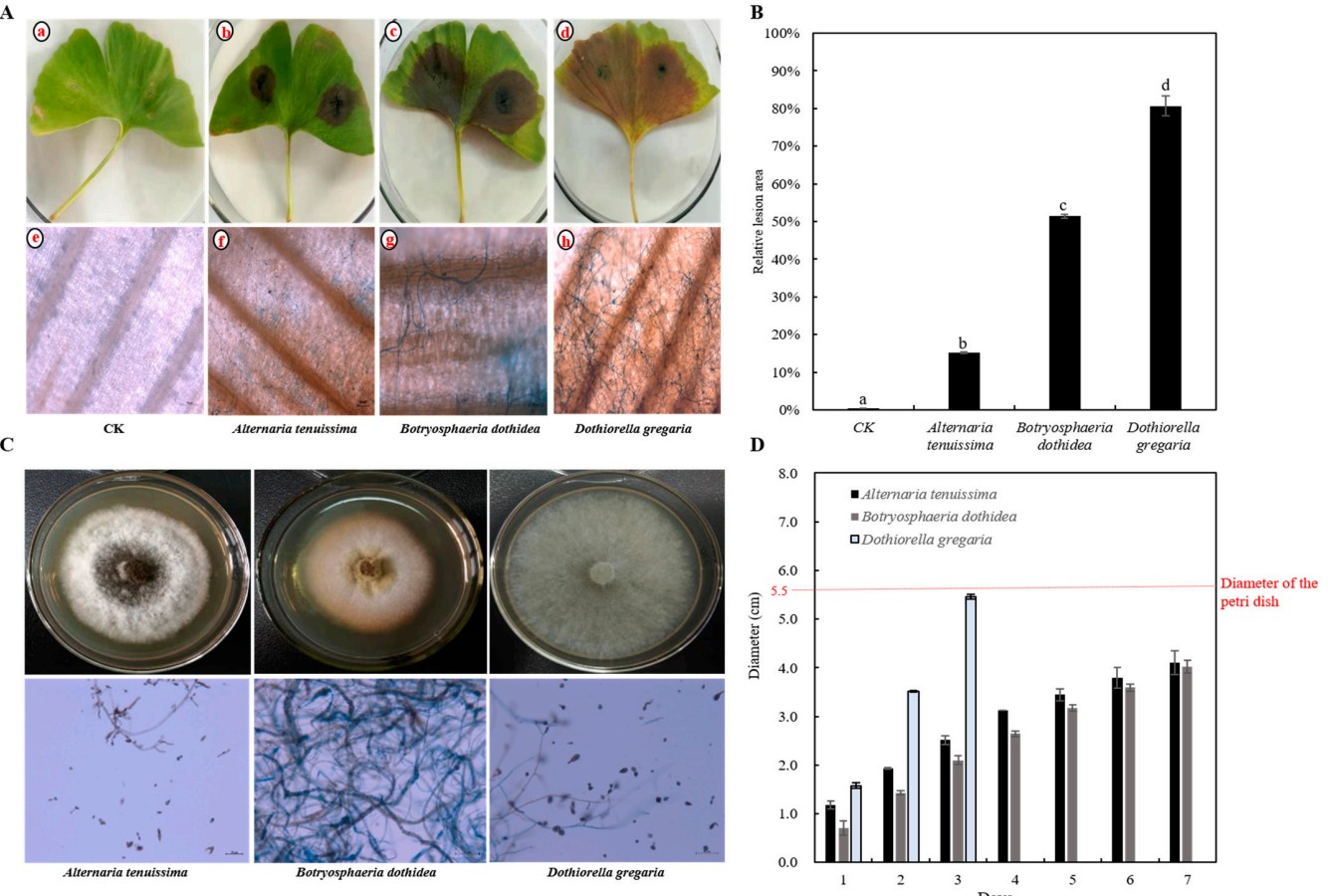

**Figure 2.** Isolation of ginkgo leaf blight pathogens. (**A**) Representative photographs of the morphologies (upper panel) and microscopic observations (lower panel) of infected leaves by candidate pathogens. Sterile ginkgo leaves were wounded and inoculated with candidate pathogens (on PDA) for 7 days in vitro. The blank PDA medium inoculated in the wound was used as the control (CK). The infected leaves were decolorized and then stained by Coomassie brilliant blue solution for mycelium observation. (**B**) Quantification of relative leaf lesion areas in (**A**). The Image 6 software was used to calculate the relative lesion areas. Data are expressed as mean values with standard deviation (S.D.), *n* = 3. Statistically significant differences among multiple groups were tested using one-way ANOVA and Dunnett's T3 test with Brown–Forsythe and Welch values in SPSS Statistics 17.0. Different letters represent significant differences with an alpha value of 0.05. (**C**) Morphologies

of the isolated pathogens (upper panel), aerial mycelia, or spores (lower panel). The fresh mycelia of the candidate pathogens in Petri plates were stained with lactophenol cotton blue stain solution and observed using a microscope after being rinsed with water. Images of mycelia were captured using a Zeiss Axio Zoom.V16 Stereo Microscope. The scale bars for observations of the infected leaves and mycelia were 200 μm and 50 μm, respectively. (**D**) Statistical quantification of hyphal growth of candidate pathogens on PDA medium in 7 days. Data are expressed as mean values with standard deviation (S.D.), *n* = 3.

Furthermore, blue hyphae in the infected leaves were clearly observed after being stained by Coomassie brilliant blue compared to the CK (Figure 2A(e–h)). These results revealed that the growth of the candidate fungi was related with the symptoms of infection points. The quantitative results showed that the relative lesion areas in the groups co-cultured with the candidate pathogens were significantly larger than those of the CK (Figure 2B). Moreover, the lesion areas caused by three candidate pathogens showed significant differences, among which the lesion area caused by *Dothiorella gregaria* was the largest (Figure 2B). Finally, the three fungi were confirmed as the primary candidate pathogens of ginkgo leaf blight in this study as they were re-isolated from the infected tissues (Figure 2C and Table S4).

The colony appearance and growth characteristics of the candidate pathogens were also monitored. The fresh mycelium of *A. tenuissima* was grey-white and then turned blackish green, and several round spores could be seen under the microscope (Figure 2C, left column). *B. dothidea* presented a yellow-white color, and robust moniliform hyphae could be seen after being stained by lactophenol cotton blue solution (Figure 2C, middle column). By contrast, the aerial hyphae of *D. gregaria* were white and spore bundles could be observed under the microscope (Figure 2C, right column). Additionally, the hyphal growth rate of *D. gregaria* was significantly faster than that of *A. tenuissima* and *B. dothidea*, and the whole Petri plate (Φ 5.5 cm) was covered in three days (Figure 2D).

Pathogenic tests were further performed on sterile tissue-cultured seedlings and ginkgo seedlings in the lab (Figures 3 and S3). Obvious black or dark brown patches appeared at the infection sites of the sterile tissue-cultured seedlings, which demonstrated the potential pathogenicity of the target strains on sterile ginkgo leaves (Figure S3). Moreover, *A. tenuissima*, *B. dothidea*, and *D. gregaria* could independently cause larger black patches (left part of each ginkgo leaf) compared to the CK (infected with fresh PDA medium without any hyphae; right part) (Figure 3A,B). In addition, the multi-site inoculation and co-infection experiments showed that the pathogens could cause significantly larger lesions compared to the CK (Figure 3C,D). Among them, *A. tenuissima* might play a bigger role, causing almost totally wilted leaves upon infection with it, followed by *B. dothidea* (Figure 3C(b,c),D).

### 3.3. In Vitro Effects of Flavonoids on the Growth of Ginkgo Leaf Blight Pathogens

The level of total flavonoids in the healthy part of the infected leaves was significantly higher than in the healthy leaves (Figure 1B). This revealed that flavonoids from plants might resist the growth of ginkgo leaf pathogens as phytoalexins to some extent. To verify this result, flavonols (quercetin, kaempferol, and isorhamnetin) and flavones (apigenin and luteolin) were added to the PDA medium and the growth rates of the pathogens were monitored. The results showed that apigenin at a concentration of 10 mg/L could significantly inhibit the growth of the three pathogens by 8.3–13.4% (Figure 4A,B). Similarly, luteolin at 10 mg/L could inhibit the growth of *A. tenuissima* and *B. dothidea* by 11.67% and 13.80%, respectively (Figure 4A,B). By contrast, kaempferol could only suppress the extension of hyphae of *D. gregaria* by 21.6% (Figure 4A,B). No obvious differences were observed for the growth of pathogens treated with other tested flavonoids (Figure 4A,B).

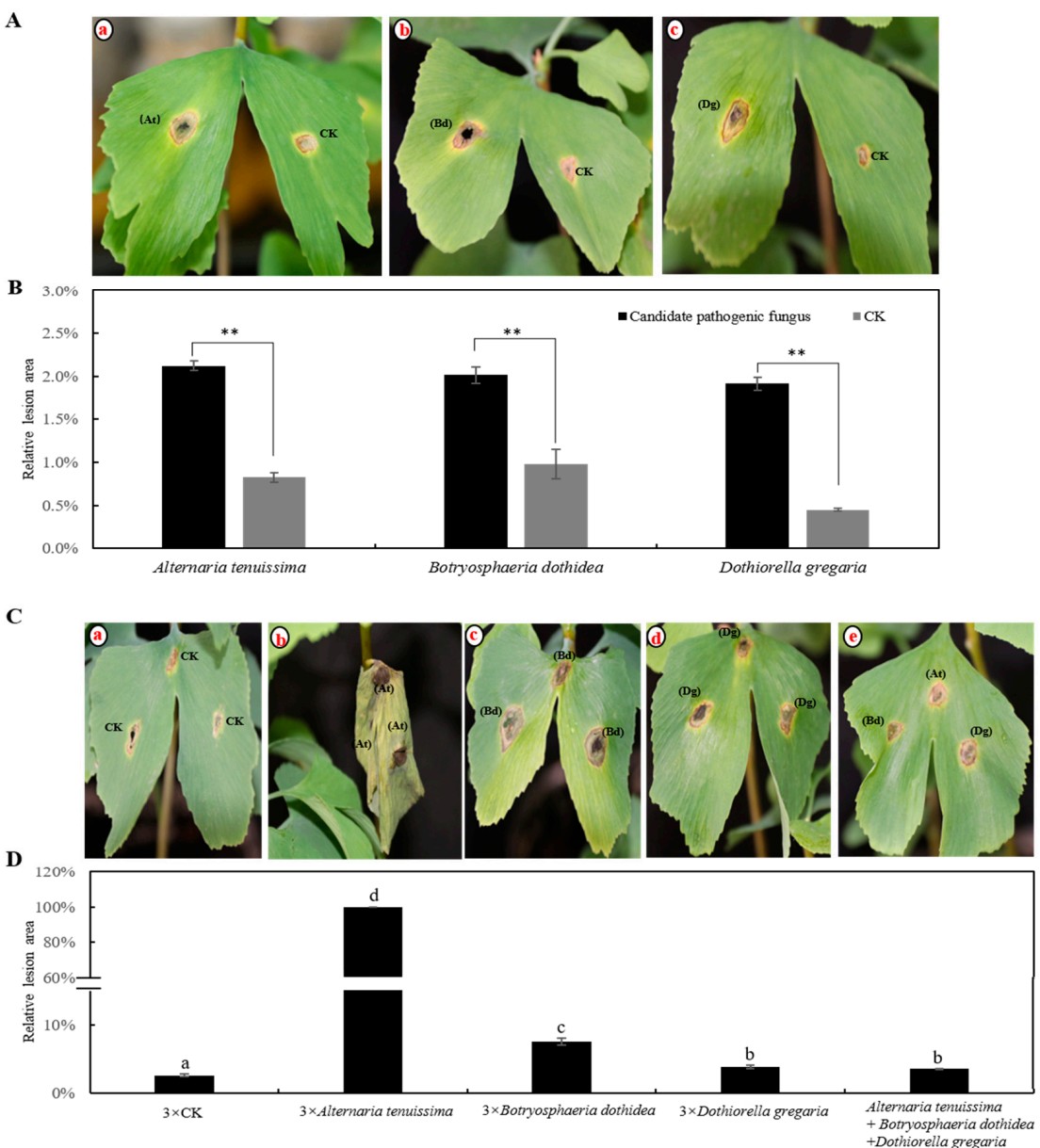

**Figure 3.** Identification of ginkgo leaf blight pathogens. (**A**) Representative photographs of the morphologies of leaves infected by independent candidate pathogens in the laboratory. Ginkgo leaves were wounded and inoculated with candidate pathogens (on PDA) for 14 days. The blank PDA medium inoculated in the wound was used as the control (CK). (**B**) Quantification of relative leaf lesion areas in (**A**). Image 6 software was used to calculate the relative lesion areas. Data are expressed as mean values with standard deviation (S.D.), $n = 3$. Differences between two groups were analyzed using independent samples t-test under the function of Compare Means (sig.$^{\text{two-tailed}}$ = 0.000, with equal variance; ** represents sig $^{\text{two-tailed}}$ < 0.01). (**C**) Morphologies of ginkgo seedlings infected via multi-site inoculation and co-infection of candidate pathogens for 14 days. (**D**) Quantification of relative lesion areas in C. The Image 6 software was used to calculate the relative lesion areas. Data are expressed as the mean values of the sum of three lesions with standard deviation (S.D.), $n = 3$. Statistically significant differences among the multiple groups were tested using a one-way ANOVA with Dunnett's T3 test in SPSS Statistics 17.0. Different letters represent significant differences at an alpha value of 0.01. At, *Alternaria tenuissima*; Bd, *Botryosphaeria dothidea*; Dg, *Dothiorella gregaria*; Ck, Control.

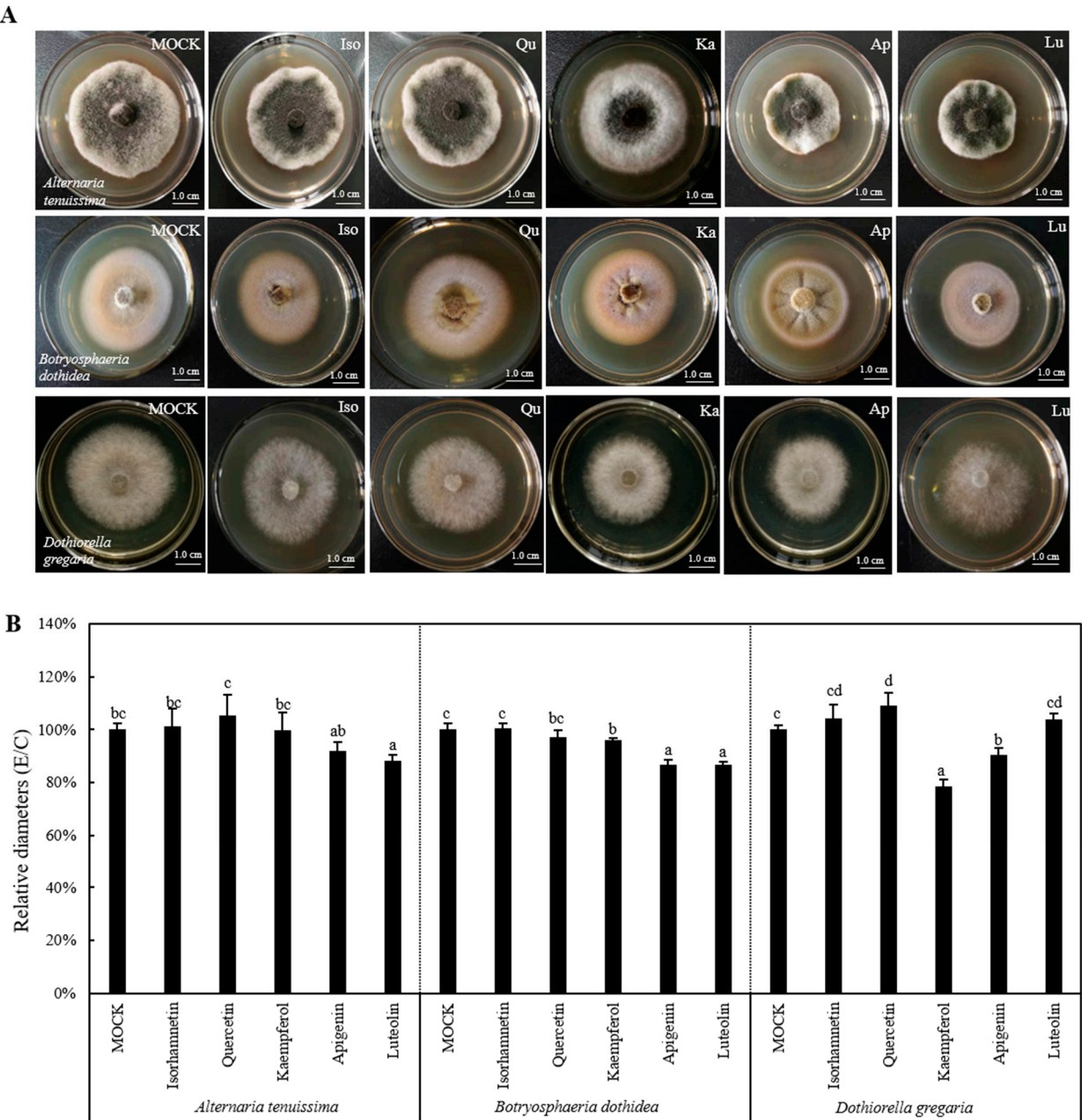

**Figure 4.** Morphology observation (**A**) and quantification (**B**) of leaf blight pathogens' hyphal growth rate on PDA medium treated with different types of flavonoids at a concentration of 10 mg/L. The colony statuses are presented on day 5 for *Alternaria tenuissima*, on day 4 for *Botryosphaeria dothidea*, and on day 2 for *Dothiorella gregaria*. Statistically significant differences were tested using one-way ANOVA with LSD (L) and Duncan (D) test in SPSS Statistics 17.0. Different letters represent significant differences at an alpha value of 0.05.

### 3.4. Isolation, Identification, and Phylogenetic Analysis of Ginkgo Leaf Endophytic Fungi

In this study, fifty-four leaves from the lateral branches in the crown of nine healthy trees (6 leaves per tree) at three sites were sampled and used to isolate endophytic fungi. In total, fifty-six internal transcribed spacer (ITS) sequences were obtained and identified as specific fungi with over 95% similarity when compared to the reference sequences in the NCBI database (Table S2; Figure S4). The identified endophytic fungi were distributed in 14 genera, including *Alternaria* sp. (25), *Phyllosticta* sp. (11), *Chaetomium* sp. (5), *Colletotrichum* sp. (3), *Dothideomycetes* sp. (2), and *Fusarium* sp. (1). Among them, the genus *Alternaria* was the

most widely represented, particularly the species *A. alternata* (10) and *A. tenuissima* (6) (Table S2; Figure 5A).

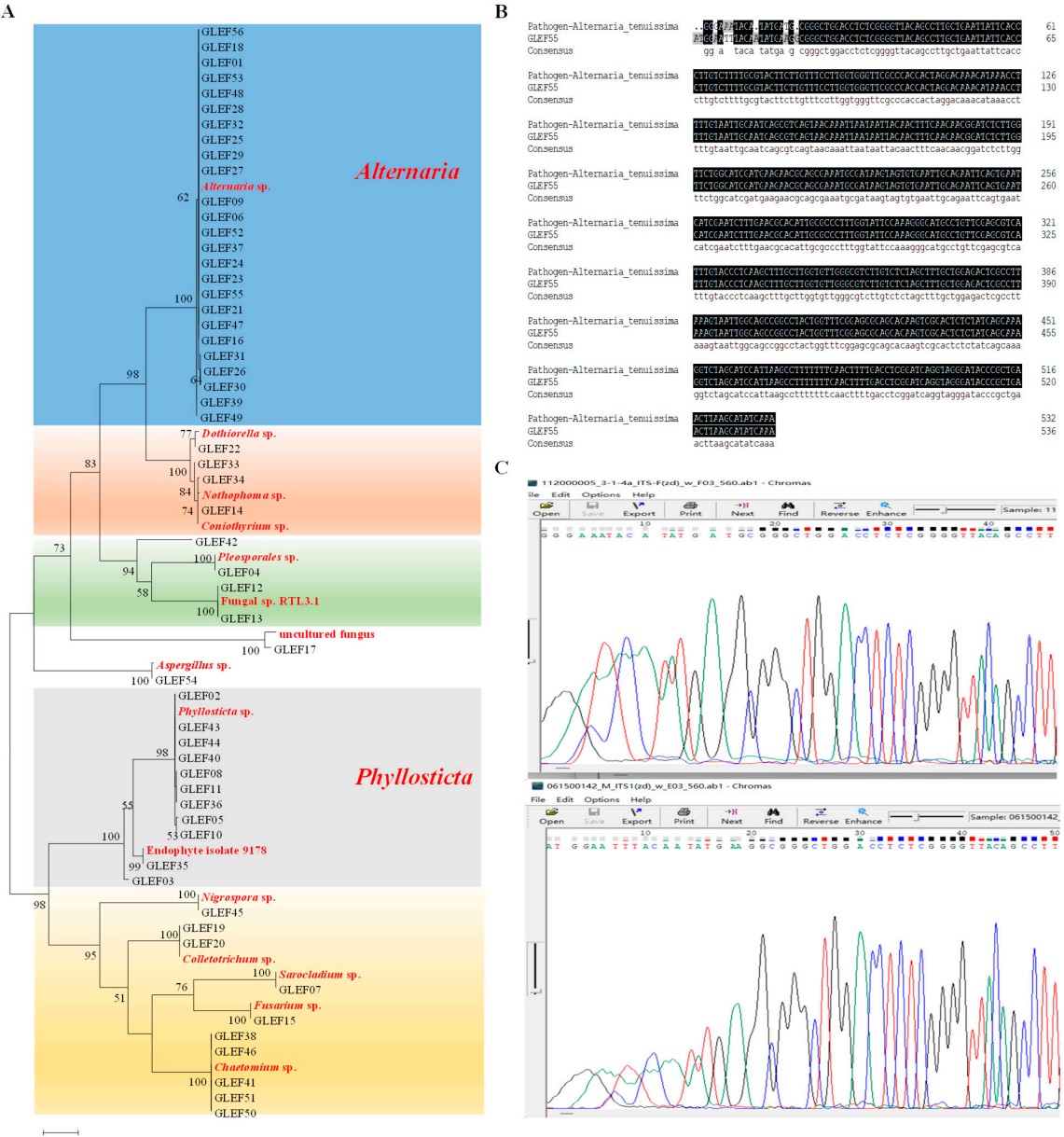

**Figure 5.** Internal transcribed spacer (ITS) sequence analysis of identified fungi. (**A**) Phylogenetic analyses of the ITS sequences of endophytic fungi isolated from ginkgo leaves. Multiple sequences were aligned using Clustal W (https://www.genome.jp/tools-bin/clustalw, accessed on 20 March 2023). A phylogenetic tree based on maximum likelihood, the Tamura–Nei model, and the nearest-neighbor interchange (NNI) ML heuristic method under complete deletion gaps and 1000 replicates of bootstrap replications was constructed using Mega 6.0. Colorful shadows represent the different genera of the isolated fungi. The GenBank accession numbers for the ITS sequences of ginkgo leaf endophytic fungi (GLEF) are listed in Table S2. The accession numbers for the Genbank reference ITS sequences are listed in Table S3. (**B**) ITS sequence alignments of the endophytic fungus GLEF55 and the ginkgo leaf blight pathogen *A. tenuissima* using DNAMAN 8.0. (**C**) ITS sequencing chromatograms of the pathogen *A. tenuissima* (upper panel) and GLEF55 (lower panel) using the CHROMAS software (version 2.6.5). Colored peaks represent different deoxynucleotides, and green for "A", black for "G", blue for "C" and red for "T".

The phylogenetic analysis showed that 25 *Alternaria* members were grouped into one big cluster with the reference fungus *Alternaria* sp. (Figure 5A, blue part). *Dothiorella* clustered with GLEF22 (*Dothiorella* sp.), GLEF33 (*Dothiorella gregaria* isolate BJ17), GLEF34 (*Didymellaceae* sp.), and GLEF14 (*Peyronellaea* sp.) and formed another clade neighboring with *Alternaria* (Figure 5A, brown part). In addition, 11 *Phyllosticta* spp. fungi clustered with the endophyte isolate 9178 and the reference fungus *Phyllosticta* sp., forming the second largest independent genus cluster (Figure 5A, grey part). Moreover, GLEF45 (*Nigrospora* sp.) clustered with *Colletotrichum*, *Sarocladium* sp., *Fusarium* sp., and *Chaetomium* sp., independently forming another large clade neighboring the clade of *Phyllosticta* (Figure 5A, yellow part). Notably, GLEF17, an uncultured soil fungus, was branched independently and close to *Aspergillus ruber* GLEF54 (Figure 5A, white part). Other minor strains clustered together and formed a loose branch, which revealed their relatively distant evolutionary status with the *Alternaria* and *Phyllosticta* clusters (Figure 5A, green part).

To explore the potential relationships between the ginkgo leaf endophytic fungi and ginkgo leaf blight pathogens, the phylogenetic analysis of their ITS sequences was conducted. The results showed that *A. tenuissima* clustered in the biggest clade of *Alternaria* and was the closest to GLEF55 (*A. tenuissima*) (Figure S5). This revealed the very close evolutionary relationship between GLEF55 and the pathogen *A. tenuissima*. Further analysis showed that the endophytic fungus GLEF55 had the same ITS sequence as the ginkgo leaf blight pathogen *A. tenuissima*, which was confirmed by the multiple sequence alignments and ITS sequencing chromatograms (Figure 5B,C).

*3.5. Metabolite Analysis of Endophytic Fungi and Their Effects on the Growth of Ginkgo Leaf Blight Pathogens In Vitro*

Flavonoid-producing endophytic fungi were selected through fermentation in a YES (yeast-extract sucrose) liquid medium. Among the fifty-six isolated endophytic fungi, forty showed an obvious proliferation of mycelia, and the total flavonoid contents in the fermentation broth were 0–90.19 mg/L, as determined by chemical colorimetric methods (Figure S6A). Four fungi (*Alternaria* sp., GLEF39; *A. tenuissima*, GLEF55; uncultured soil fungus, GLEF17; and *Aspergillus ruber*, GLEF54) were proven to accumulate relatively higher levels of flavonoid-related compounds (Figure 6A,B).

Among them, GLEF39 and GLEF55 both belonged to the *Alternaria* cluster, and were different in colony appearances (Figure 6A). The GLEF39 colony appeared white, and the color was dark greyish green for GLEF55. By contrast, the GLEF17 and GLEF54 colonies were yellow (Figure 6A). The quantitative analysis showed that the total flavonoid content of the fermentation broth was around 20.26–79.86 mg/L (Figure 6B). Further analysis showed that the flavonoid profiles in the isolated fungi were quite different from those in the ginkgo leaves (Figure S6B). Flavonol glycosides in the ginkgo leaves were mostly distributed within the retention time points of 8–20 min in our analysis, which were the major parts of the medicinal composition of standard ginkgo leaf extract [37]. However, the flavonoid compounds in the endophytic fungi were mainly distributed in the time points from 15 to 40 min (Figure S6B). In this study, we did not observe specific flavonoids that were present in both the ginkgo leaves and the fungi according to the UV–ultraviolet spectrum and typical MS ion fragments (Figure S6B).

Additionally, the anti-pathogen (ginkgo leaf blight) activities of the extracts of the candidate endophytic fungi were tested. The results showed that the extracts of the candidate endophytic fungi exhibited different effects on the growth of pathogens. The extract of GLEF55 could significantly inhibit the growth of *B. dothidea* and *D. gregaria* by 20.4% and 22.0%, respectively, but not the growth of *A. tenuissima* (Figure 6C,D). By contrast, GLEF39, GLEF17, and GLEF54 promoted the growth of *A. tenuissima* and *B. dothidea* but inhibited the growth of *D. gregaria* by 33.3–44.7% (Figure 6C,D).

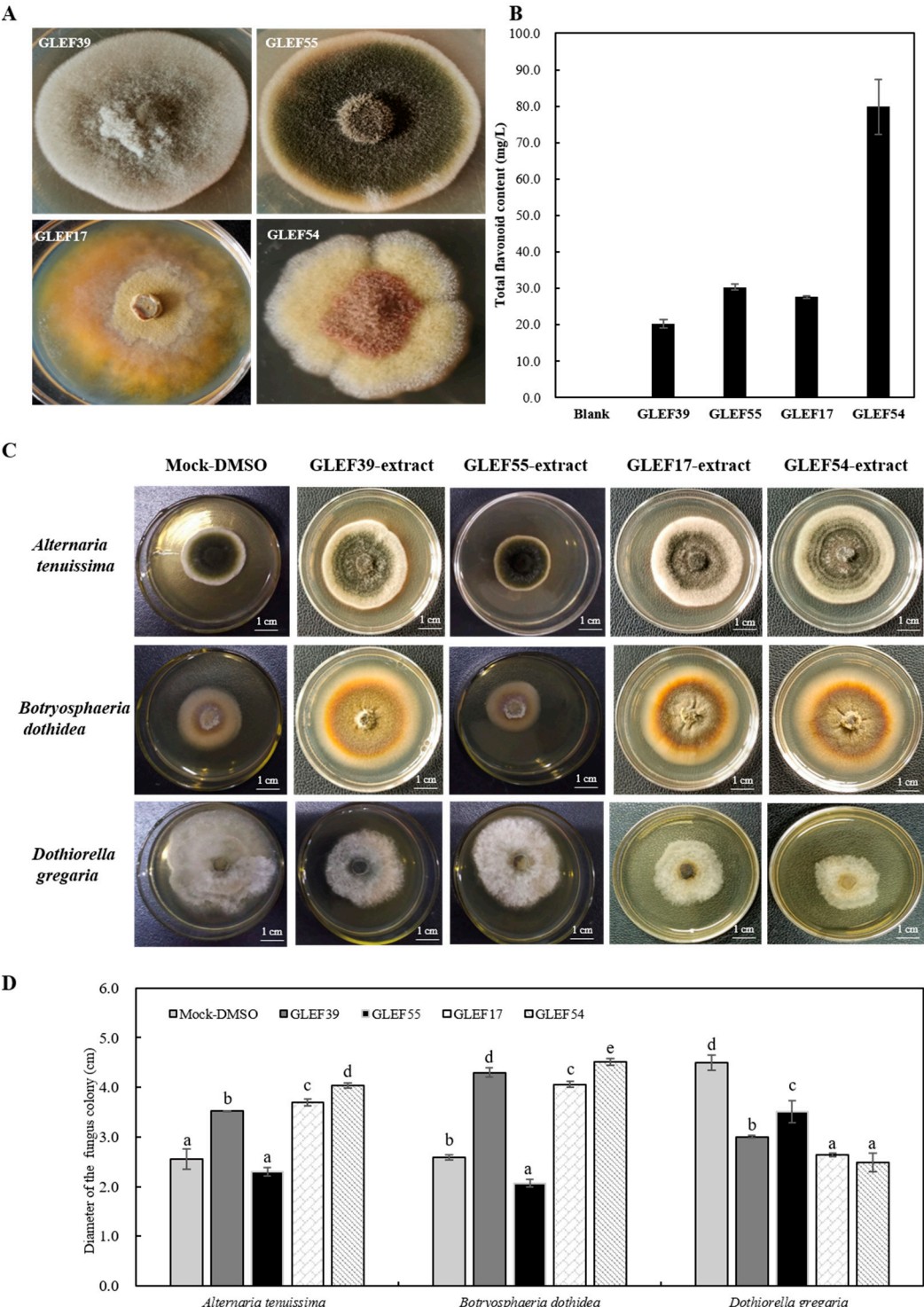

**Figure 6.** Metabolite analysis of endophytic fungi and their effects on ginkgo leaf blight pathogens in vitro. (**A**) Morphologies of the isolated flavonoid-producing ginkgo leaf endophytic fungi (GLEF). (**B**) Total flavonoid determination in the isolated GLEF in (**A**). (**C**) Morphologies of ginkgo leaf blight pathogens treated with the extracts of GLEF (the final concentration of the metabolites in the medium is 1/1000 of the original mixture in (**B**). (**D**) Quantification of the colony size by diameter. Data are expressed as the mean values of three replicates with standard deviation (S.D.). Statistically significant differences were tested using one-way ANOVA with LSD (L) and Duncan (D) test in SPSS Statistics 17.0. Different letters represent significant differences at an alpha value of 0.05.

### 3.6. Dual Culture of Endophytic Fungi and Ginkgo Leaf Blight Pathogens In Vitro and In Vivo

To investigate the potential relationships between flavonoid-producing endophytic fungi and ginkgo leaf blight pathogens, dual cultures were carried out in Petri plates. Several confrontation categories were presented according to the final distribution patterns of the fungi in the Petri plates. The endophytic fungus GLEF39 showed an obviously slower ($p = 0.012$) growth rate when co-inoculated with *A. tenuissima* (Figure 7A(a)). This confrontation situation was especially apparent ($p = 0.000$) in the dual cultures of GLEF17/GLEF54 with the pathogens (Figure 7A(g–l)). No advantages of the slow-growing endophyte GLEF17 were observed over the rapid-growing pathogens, which could be partially reversed by the endophytic fungus GLEF17 during the period of vigorous growth (Figures 7A(g–i) and S7A). Interestingly, a visible inhibition zone appeared even though the endophytic fungus GLEF54 was not dominant in growth when faced with the pathogens (Figures 7A(i,j) and S7B).

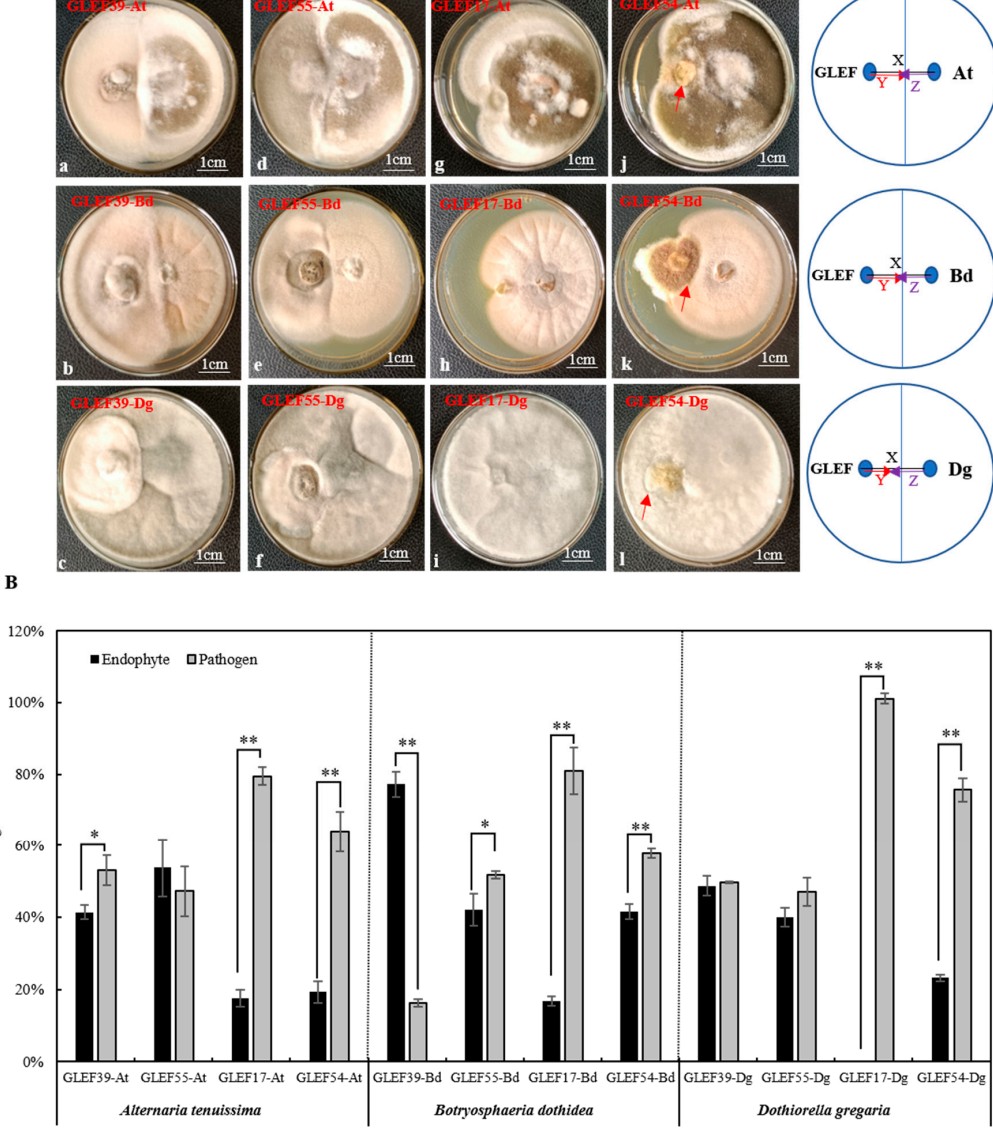

**Figure 7.** Dual cultures of ginkgo leaf endophytic fungi (GLEF) and ginkgo leaf blight pathogens in vitro. (**A**) Dual cultures of endophytic fungi and ginkgo leaf blight pathogens in vitro. The pathogens and GLEF were simultaneously inoculated into the Petri plates in opposite positions. The simulation diagrams of the dual culture are represented on the far right. For each dual culture, endophytic fungus

is placed on the left side in Petri plate, and the pathogen is on the right. (**B**) Quantification of the dual cultures based on relative growth rate. The relative growth rate was calculated as follows: the relative growth of endophyte = Y/X × 100%; the relative growth of pathogen = Z/X × 100%; X represents the distance between the center points of an endophyte and a pathogen; Y represents the distance between the center of the endophytic fungus to its colony edge along the center line; and Z represents the distance between the center of the pathogenic fungus to its colony edge along the center line. The differences between two groups were analyzed using an independent samples t-test under the function of compare means (** represents sig $^{\text{two-tailed}}$ < 0.01; * represents sig $^{\text{two-tailed}}$ < 0.05). Data are expressed as the mean values of three replicates with standard deviation (S.D.), *n* = 3.

By contrast, GLEF39 (*Alternaria* sp.) showed the significant ($p$ = 0.000) growth advantages over *B. dothidea* (Figure 7A(b),B). GLEF55 presented the similar ($p$ > 0.05) growth rate as *A. tenuissima* (Figure 7A(d),B). Although GLEF39 and GLEF55 showed similar ($p$ > 0.05) growth stage as *D. gregaria*, a clear boundary existed (Figure 7A(c,f),B). These results revealed that GLEF39 could exist independently during the invasion of pathogens, which was also applicable to GLEF55 (Figure 7A(d–f),B). As mentioned above, the endophytic fungus GLEF55 had the same ITS sequence and similar colony morphology as the pathogen *A. tenuissima* (Figures 2 and 5). Therefore, they were considered to be the same, which was further proven by their nip and tuck growth speed in the dual culture (Figure 7A(d)).

Moreover, the candidate endophytic fungi and pathogens were symmetrically inoculated into living ginkgo leaves. The results showed that the lesion areas of the ginkgo leaves at the side where the endophytic fungus GLEF39 or GLEF54 and specific pathogens were both inoculated were significantly ($p$ < 0.05 or 0.01) larger and the leaves exhibited more obvious wilting phenotypes than the control (Figure 8A(a,e,i,d,h,l),B). These results demonstrated that the endophytic fungi GLEF39 and GLEF54 also exhibited additional pathogenicity in *G. biloba* leaves when inoculated with pathogens. GLEF55 could cause similar lesions on leaves when co-inoculating with *A. tenuissima* in ginkgo (Figure 8A(b)). However, no significant differences ($p$ > 0.05) were observed when GLEF55 was inoculated with the three pathogens compared to the control (Figure 8A(b,f,j),B). These results showed the variable roles of GLEF55 in the ginkgo leaf blight. In addition, the endophytic fungus GLEF17 might play staggered roles when facing the pathogens. The pathogenicity of *A. tenuissima* and *D. gregaria* could be significantly ($p$ = 0.001 and 0.000) counteracted by GLEF17 to some extent (Figure 8A(c,k),B). However, GLEF17 could also dramatically enhance ($p$ = 0.000) the pathogenicity of *B. dothidea* (Figure 8A(g),B). These results revealed the multiple roles that these endophytic fungi played in the occurrence of ginkgo leaf blight.

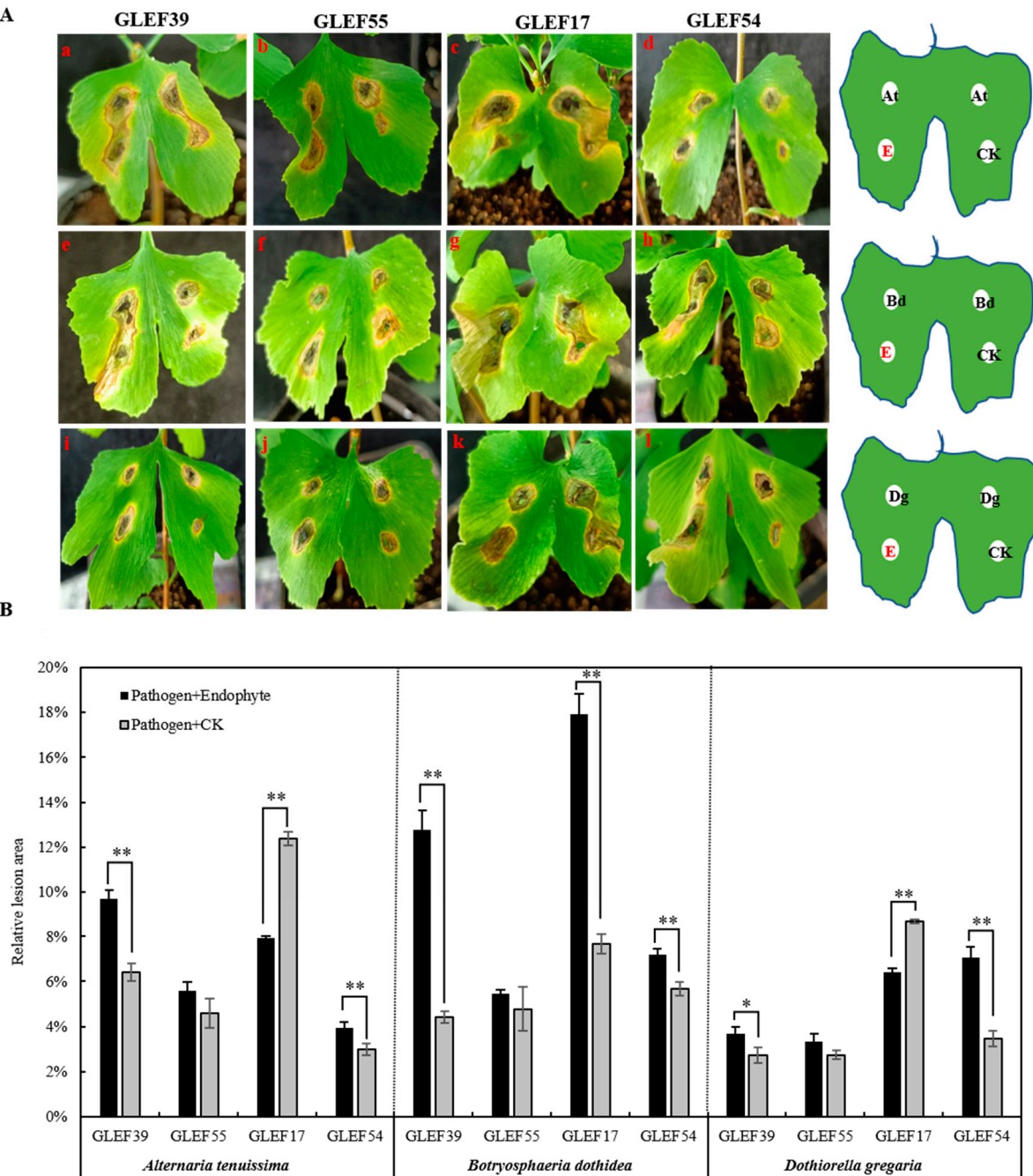

**Figure 8.** Dual cultures of ginkgo leaf endophytic fungi (GLEF) and ginkgo leaf blight pathogens in vivo. (**A**) Dual cultures of endophytic fungi and ginkgo leaf blight pathogens in vivo. The pathogens and GLEF were simultaneously inoculated in ginkgo seedlings in the lab in opposite positions of the leaves and were then cultured for 14 days. The simulation diagrams of the dual culture are represented on the far right. At, *Alternaria tenuissima*; Bd, *Botryosphaeria dothidea*; Dg, *Dothiorella gregaria*; CK, control. (**B**) Quantification of the dual culture based on relative lesion areas in ginkgo leaves. The Image 6 software was used to calculate the relative lesion areas. Data are expressed as mean values with standard deviation (S.D.), $n = 3$. Differences between two groups were analyzed using independent samples t-test under the function of Compare Means (** represents sig $^{\text{two-tailed}} < 0.01$; * represents sig $^{\text{two-tailed}} < 0.05$).

## 4. Discussion

The relationships among host plants' metabolites (flavonoids), endophytic fungi, and ginkgo leaf blight pathogens remain largely unknown. Therefore, we investigated the flavonoid profiles of infected ginkgo leaves, pathogens of ginkgo leaf blight, and endophytes of ginkgo and the potential relationships among them. Our data integrate a long debate on the causes of ginkgo leaf blight and offer new insights into the relationships between plant pathogens and endophytes in ginkgo leaf.

### 4.1. Flavonoids and Their Role in Plant Resistance to Fungi

Flavonoids, as a major group of secondary metabolites in plants, play essential roles in protecting plants against pathogen infection as phytoalexins [38,39]. The co-expression of *PalbHLH1* and *PalMYB90* from *Populus alba* increases the flavonoid content and enhances the pathogen resistance of poplar to *Dothiorella gregaria* Sacc and *Botrytis cinerea* [40]. In addition, accumulations of specific flavonoids (e.g., 3-deoxyanthocyanidins, flavan-3-ols, and O-methylflavonoids) after infection with pathogenic fungi also play important roles in disease resistance in several plants, such as sorghum (*Sorghum bicolor*), poplar, and maize (*Zea mays*) [39,41–44].

In this study, the levels of total flavonoids in the healthy–sick (H-S) parts of the ginkgo leaf blight-infected leaves increased compared to the healthy leaves (H), while the levels in sick parts (S) decreased (Figure 1A,B). This suggested that ginkgo leaves showed an induced defense response that consisted of the accumulation of some flavonoids (e.g., kaempferol) when facing pathogens, which had been reported in *Astragalus adsurgens* [45]. However, there were no obvious changes in the types of total flavonoid composition (Figure S2). Lu et al. [46] showed that total flavonoid content in ginkgo leaf blight-infected leaves slightly decreased but not significantly ($p > 0.05$). These somewhat conflicting results might be attributed to the different measuring methods used. In the previous study, the authors did not distinguish sick parts and H-S parts, but we did in this study (Figure 1B). In addition, kaempferol, apigenin, and luteolin at a concentration of 10 mg/L could significantly suppress the growth of pathogens by 8.3–21.6% in vitro (Figure 3). Flavones (e.g., apigenin) had been proven to have fungicidal activity toward some agricultural pathogens at 3–50% [47]. These works suggested that flavonoids could inhibit the elongation of pathogenic hyphae and potentially delay the onset time of the disease to some extent.

Compounds originating from endophytic fungi are also an alternative source of chemical with fungicidal or fungistatic effects [48]. In this study, the flavonoid profiles in the endophytes were quite different compared to those in the natural ginkgo leaves (Figure S6B). Although many antibacterial substances (e.g., apigenin-8-C-β-D-glucopyranoside) had been isolated and identified from the endophytes of ginkgo, the progress in the search for similar pharmacodynamic substances in ginkgo leaf is extremely limited [26,49]. Moreover, weak anti-pathogenic properties of the tested endophytic extracts were found in this study (Figure 6). This suggested that it might be challenging to explore antifungal endogenous metabolites through the tested endophytic fungi in this study. However, at the very least, some endophytic fungi (GLEF17 or GLEF54) might be promising biocontrol candidates for ginkgo leaf blight (Figures 7 and S7).

### 4.2. Potential Relationships between Leaf Endophytes and Ginkgo Leaf Blight
4.2.1. Debate on the Etiology of Ginkgo Leaf Blight Disease

*Ginkgo biloba*, the sole surviving member of the Ginkgoaceae family, has been called a "living fossil". It has been used worldwide due to its tremendous value in the pharmaceutical and cosmetic industries, health food, and landscape gardening [36,50]. Ginkgo leaves play an essential role in the ginkgo-related industries owing to ginkgo-specific pharmacological chemicals such as terpene trilactones (ginkgolides) and flavonol glycosides [50]. Although ginkgo is considered to be highly resistant to biotic and abiotic stresses, leaf blight disease has been very prevalent in many places of China [1–8]. In this study, the

disease was found in many places in the Xinxiang urban district (Henan province, China) (Figure S1), and the trends continued to increase.

As a destructive foliar disease of *G. biloba*, ginkgo leaf blight typically imposes a greater challenge on the survival of ginkgo [1]. There are two main arguments about the pathogenesis of ginkgo leaf blight in China. One argument is that ginkgo leaf blight is the result of infection by a variety of pathogens, including *Alternaria* sp., *Colletotrichum* sp., *Gloeosporium* sp., *Phyllosticta* sp., *Fusarium* sp., and *Pestalotia* sp. Zhu and You isolated one fungus or more from infected leaves [7,8,10] and demonstrated their pathogenic role in the emergence of ginkgo leaf blight by means of stab or wound inoculation [7,10]. However, no further details were provided about the pathogenicity of ginkgo leaf blight pathogens.

In this study, three fungi (*Alternaria tenuissima*, *Botryosphaeria dothidea*, and *Dothiorella gregaria*) were identified as the pathogens of ginkgo leaf blight in Xinxiang by identifying their pathogenicity in vitro and in vivo (Figures 2, 3 and S3). *A. tenuissima*, belonging to the *Alternaria* sp., is considered a leaf blight or leaf spot pathogen of many plants, such as *Senna nomame* and *Luffa cylindrica* [51,52]. *B. dothidea* and *D. gregaria* were first isolated and characterized as the pathogens of ginkgo leaf blight. *B. dothidea*, considered a latent pathogen of global importance to woody plant health, is among the most widespread pathogens of trees worldwide [53,54]. *D. gregaria* is also a pathogen of woody plants, such as poplar [40]. The diversity of ginkgo leaf blight pathogens might be attributed to the influence of geographical regions, which is also common in other plant pathogens (e.g., *Ceratocystis manginecans*) [55].

Additionally, some researchers suggested that foliar fungal pathogens may adversely affect an entire plant and are closely connected to environmental speciation. Thus, it is necessary to detect and identify them early in the infection process to mitigate the injury caused by these diseases [56]. Therefore, research related to ginkgo leaf blight prevention has been carried out [2,11,57–60]. These authors reported that a mixture of propiconazole at $1.25 \times 10^4$ g/mL Rovral at $5 \times 10^4$ g/mL and thiophanatemethy at $7 \times 10^4$ g/mL could reduce the incidence rate of ginkgo leaf blight by 85% as early as mid-April in field trials [11]. Chen [7] reported that hexaconazole could be effective, with an EC50 of 26.1567 μg/mL on the pathogen *A. alternata*.

However, Nie et al. [61] were not able to isolate pathogenic fungus from infected leaves. Thus, they believed that scorched leaves in Beijing had no relationship with pathogens. This might be attributed to the way in which candidate pathogens were injected into ginkgo leaves. Zhu and Shi [10] injected spores or hyphae through a wound (stabs or scalds) on the leaf. However, Nie et al. [61] inoculated ginkgo stems or leaves without puncture wounds. Chen [7] proved that wounds were essential in the identification of ginkgo leaf blight pathogens.

Correspondingly, some researchers suggested that the decreased immunity of gingko trees caused by improper cultivation and management is the main reason for ginkgo leaf blight. Therefore, they called for more attention to be placed on the improvement of cultivation conditions [5,58,60–62].

Overall, previous studies have shown that ginkgo leaf blight pathogens vary, but *Alternaria* sp. is considered the main pathogen [8–10]. Although fungicides could alleviate the course of leaf blight to some extent, they could not cure it. Furthermore, the weakened immune system of ginkgo plants caused by poor cultivation conditions might be an important reason that makes plants more susceptible to pathogens. This raises another question: where do latent pathogens (e.g., *A. tenuissima*) probably come from?

### 4.2.2. Ginkgo Leaf Endophytes and Pathogens

There are over 19,000 fungi worldwide that are known plants pathogens. Certain fungi may remain dormant but alive and may develop inside host plant tissues until conditions are conducive to their proliferation [56,63]. In contrast, endophytic fungi reside in the internal tissues of living plants but do not cause immediate, overt, negative effects on the

host plants [14]. Furthermore, some of these endophytes might be beneficial to the growth and development of the host plants, such as mycorrhizae [64].

Endophytic fungi have attracted the attention of many scientists, including botanists, chemists, pharmacologists, ecologists, mycologists, and plant pathologists, due to their specific chemicals and potential functions in environments [25,65–68]. Endophytes exist in almost 300,000 plants [69]. Several genera of endophytic fungi were isolated and identified from ginkgo, among which *Chaetomium* was found to be the most abundant in ginkgo leaves collected from Linyi (Shandong province, China), followed by *Aspergillus*, *Alternaria*, and *Penicillium* [26]. Among the isolated endophytic fungi in this study, the genera of *Alternaria* sp. (25), *Phyllosticta* sp. (9), and *Chaetomium* sp. (5) were the majority (Table S2). The differences in fungal composition might be attributed to the host genotypes, different environments, and time of sampling, which had been proven to affect the genera of epiphytic fungi in *G. biloba* and tomato [70,71].

A balanced symbiotic continuum ranging from mutualism to commensalism to parasitism exists between endophytes and host plants [12,72]. Many hostile conditions, including malnutrition, biotic stresses, and even senescence, could break the balance and lead to the transition of endophytes from the latent mode to the active virulent pathogenic mode [15,73]. For example, the fungus *Verticillium dahliae* causes wilts as a pathogen in potatoes and mint, but it also colonizes as an endophyte in mustards and grasses [74]. These findings suggest a delicate and balanced relationship between plants and their endophytic fungi. In this study, *A. tenuissima*, *B. dothidea*, and *D. gregaria* were characterized as ginkgo leaf blight pathogens, of which *A. tenuissima* belonging to the *Alternaria* genus was also the most abundant ginkgo endophyte (Figures 3 and 5A; Table S2). The endophytic fungus GLEF55 (accession number: OQ591986) was identified as *A. tenuissima*, with the same ITS sequence, colony appearance, and pathogenicity as the major pathogen *A. tenuissima*. These results suggested that the endophyte GLEF55 (*A. tenuissima*) might be a latent leaf blight pathogen in ginkgo plants. Additionally, *Colletotrichum* sp., *Phyllosticta* sp., and *Fusarium* sp., considered as ginkgo leaf blight pathogens in Nanjing (Jiangsu Province, China), were also present as endophytes in this study (Figure 5) [8]. These results suggested that it might be unavailing to try to clearly distinguish endophytes and plant pathogens as they share similar living spaces and multiple roles in plant functioning. This study revealed that some endophytes were disguised pathogens of leaf blight in ginkgo leaves, among which *A. tenuissima* was the major pathogen. This could be illustrated by their similar pathogenicity when injected into ginkgo leaves (Figure 8A).

In conclusion, the balanced symbiotic continuum of endophytes and ginkgo leaf blight pathogens can be kept when gingko trees are in a good condition or under stress that the trees could overcome by regulating the accumulation of stress-response metabolites, such as flavonoids. Otherwise, the balanced symbiotic continuum will be destroyed under stressful conditions, including malnutrition, biotic stresses, drastic climatic changes, or senescence, and endophytic fungi may turn into active virulent pathogens (Figure 9) [15,16,75,76]. This is supported by the persistence of *Alternaria* throughout the course of the disease, the difficulty of prevention and control it, as well as the practice of ginkgo leaf blight control, in which good cultivation managements were found to be more effective than fungicides [5,8,9,58,60–62].

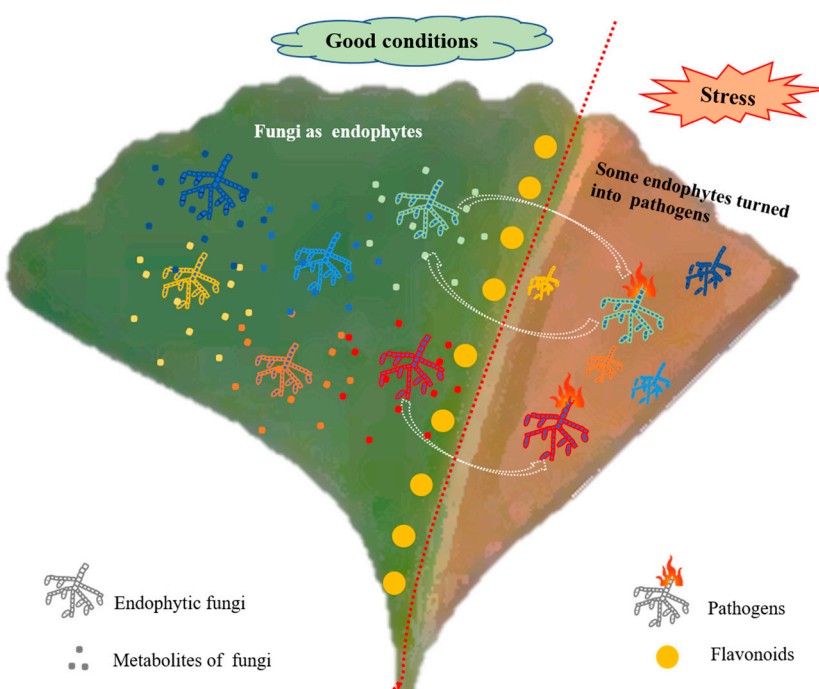

**Figure 9.** Schematic diagram of the possible relationships between endophytic fungi and ginkgo leaf blight pathogens. The balanced symbiotic continuum of endophytes and ginkgo leaf blight pathogens can be kept when plants are in good condition. Otherwise, the balanced symbiotic continuum will be destroyed under stressful conditions, including malnutrition, biotic stresses, drastic climate changes, or senescence, and some endophytic fungi may turn into active virulent pathogens. During infection, stress–response metabolites, such as flavonoids, accumulate to delay the onset time of the disease. Finally, the infected ginkgo tree suffers from leaf blight disease due to the imbalance of endophytic fungi caused by adverse environments. The leaf in green and dark brown represents the healthy and ginkgo leaf blight infected parts of the leaf, respectively. Endophytic fungi in divergent colors represent different fungi.

## 5. Conclusions

In this study, we described the delicate and balanced relationships among leaf blight pathogens, plant leaf metabolites, and endophytes in ginkgo. Flavonoids (especially flavonols) are particularly accumulated at the infection sites when pathogens are invading. Moreover, endophytic fungi present in plants could potentially be a source of unexpected pathogens, as their abundance as endophytes in healthy plants overlaps with their pathogenicity in infected ginkgo leaves under certain situations. Our conclusions can be proven by leaf blight control practice, in which good cultivation managements are more essential compared to the only use of fungicides.

**Supplementary Materials:** The following supporting information can be downloaded at https://www.mdpi.com/article/10.3390/f14071452/s1. Table S1: Basic information of the samples used for ginkgo leaf blight pathogen isolation in this study. Table S2: Endophytic fungi isolated from the ginkgo leaves in this study. Table S3: Genbank accession numbers of the reference fungi used in phylogenetic analysis. Table S4: The candidate fungi of ginkgo leaf blight pathogens and their pathogenicity analysis. Figure S1: Sampling information for ginkgo leaf blight pathogen isolation. Figure S2: HPLC chromatograms showing flavonoid accumulation profiles in healthy leaves (H), healthy parts of sick leaves (H-S), and sick leaves (S). Figure S3: Pathogenicity analysis of ginkgo leaf blight pathogens on tissue-cultured seedlings. Figure S4: Agarose gel electrophoresis results after ITS amplification by PCR for the identification of ginkgo leaf endophytic fungi. Figure S5: Phylogenetic analysis of the ITS sequences from ginkgo leaf endophytic fungi and the ginkgo leaf blight pathogen *Alternaria tenuissima*. Figure S6: Flavonoid profile determination and analysis of the isolated ginkgo leaf endophytic fungi

(GLEF). Figure S7: Dual culture of ginkgo leaf endophytic fungi and ginkgo leaf blight pathogens in vitro.

**Author Contributions:** Conceptualization and methodology, X.S., R.S., X.L., and M.Z.; resources, Z.Y., L.H., and H.H.; writing—original draft preparation, X.S.; writing—review and editing, X.S. and C.L.; supervision and funding acquisition, C.L. and J.C. All authors have read and agreed to the published version of the manuscript.

**Funding:** This research was funded by the National Nature Science Foundation of China, grant number 32101568 and the Scientific and Technological Research Project of Henan Province, China, with grant numbers 202102310194 and 222102110441.

**Data Availability Statement:** All data presented in this article are available upon request.

**Acknowledgments:** We thank Feng Zhou from the College of Resources & Environmental Science, Henan Institute of Science and Technology, for his kind help with the fungal identification analysis. We also thank Jialing Cai, Runge Yang, Yue Cao, Linlin Yao, Ji Chen, Yuqing Huo, and Zhaoyang Yue from the Henan Institute of Science and Technology for their kind help with the fungal metabolite analysis.

**Conflicts of Interest:** The authors declare no conflict of interest.

## Abbreviations

| | |
|---|---|
| GLEF | Ginkgo leaf endophytic fungi |
| PDA | Potato dextrose agar medium |
| ITS | Internal transcribed spacer |
| YES | Yeast-extract sucrose medium |
| HPLC-MS | High-performance liquid chromatography–mass spectrometry |
| DMSO | Dimethyl sulfoxide |
| NCBI | National Center for Biotechnology Information |

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
