# Peer review of "Friend or Foe? The Endophytic Fungus Alternaria tenuissima Might Be a Major Latent Pathogen Involved in Ginkgo Leaf Blight"

_forests, doi:10.3390/f14071452_

Round 1
Reviewer 1 Report (New Reviewer)
Section 2.6. The information concerning DNA purification is insufficient, since the article number [30] is not written in English. I would recommend adding a picture displaying the results of PCR analysis with positive/negative controls and molecular markers, at least in the Supplementary materials. The authors emphasize that “one endophytic fungus named GLEF55 obtains the same ITS sequence with ginkgo leaf blight pathogen A. tenuissima” (lines 368-369). One should keep on mind that with PCR, artifact results cannot be excluded. What was the sequencing method?
Section 3.4 and Figure 5. Bootstrap values are missing (greater than 50% or maybe more?) as well as the scale bar.
In general, the manuscript is written clearly and in good English.
Some editorial notes:
Line 535: Should be “blight” instead of “bilght”
Lines 590-591: This sentence sounds a bit strange to me. Please reformulate it.
Author Response
Please see the attachment.

Reviewer 2 Report (New Reviewer)
Dear Authors,
I thoroughly reviewed the Manuscript and commented some points. This manuscript is interesting, and I would like to recommend to improve more.
Please, see my comments in attached file.
Thanks

Authors should check more.
Author Response
Please see the attachment.

This manuscript is a resubmission of an earlier submission. The following is a list of the peer review reports and author responses from that submission.
Round 1
Reviewer 1 Report
The study aimed to investigate the causes of ginkgo leaf blight, a major disease affecting ginkgo trees in China. The researchers characterized the flavonoid accumulation profiles in infected leaves, pathogens, and endophytes of healthy ginkgo leaves. They identified three pathogens, Alternaria tenuissima, Botryosphaeria dothidea, and Dothiorella gregaria, as the cause of the disease according to Koch's postulates. They also isolated and characterized 56 endophytic fungi, including A. tenuissima. Flavonoid-producing endophytes were selected and tested for their effects on the growth rates of pathogens. The study found that A. tenuissima, a fungus found in healthy leaves, might be the main cause of ginkgo leaf blight. The researchers suggested that good cultivation practices would be more effective than fungicides for controlling the disease. Overall, the study provides valuable insights into the etiology of ginkgo leaf blight and may contribute to its management and control.
Following revision could be considered
Line no. 106: the leaves were immediately sealed independently and quickly TRANSPORTED for pathogens isolation.
Line no. 112: The healthy leaves were USED as controls.
Line no. 120: microscopy resolutions (example 400X) could be mentioned here.
Line no. 125: "and inoculated for 1-2 weeks as described". Is it every day inoculation up to 2 weeks? This line could be more clear.
Line no. 132: "15 minutes for surface sterilized" can be "15 minutes for surface sterilization"
Line no. 140: "Fungi were identified with over 95% identities of ITS sequences with the data in" can be "Fungi were identified with over 95% similarity of ITS sequences with that of the reference data in".
Line no. 181: The first line could be removed. No need for citations in result section.
Line no. 246: "higher than in the healthy" can be "higher than in the healthy leaves".
Line no. 292: Fig. 4 could be re-drawn by including the Genbank refernce sequences in the phylogram.
Line no. 516: Can be re-written for better clarity. example: "Moreover, the endophytic fungi present in plants could potentially be the source of unexpected pathogens, as their abundance as endophytes in healthy plants overlaps with their pathogenicity to ginkgo leaves in certain situations."
Author Response
Please see the attachment, thank you.

Reviewer 2 Report
Comments
1. In abstract section, line 28 the authors need to add petriplates or petridishes instead of plates.
2. In line 31, pathogens were inoculated into……the sentence looks incomplete. Please correct the sentence.
3. In introduction section, line 53 needs spacing within words and also correct the species abbreviation (sp.)
4. In materials and methods section, line 208 use italics (eg. in vitro)
5. In line 209 and 210, the author needs to italicize the fungus names.
6. In fig 3, the author needs to italicize the fungus names within the graph.
Author Response
Please see the attachment, thank you.

Reviewer 3 Report
The manuscript determined the flavonoid accumulation profiles in infected leaves, pathogens and endophytic fungi of healthy leaves. In addition, co-inoculations of endophytes and ginkgo leaf blight pathogens on ginkgo were also evaluated. Overall, the study is comprehensive, and the work has been performed in an acceptable manner. However, it would be helpful to the readers if the authors could consider the following points.
Introduction
L87-88: “The present study tried to explore it”. Suggest being more specific. What were the authors exploring? The interactions between the hosts, pathogens and endophytic fungi, but in what aspect? Or in vitro or in vivo?
L100: “Pathogens isolation” change to “Pathogen isolation”.
L105: Until 12 am?
L106: “quickly for pathogen isolation”. Perhaps it can be revised to “quickly brought back to the laboratory for pathogen isolation”.
L108 & 125: “Chen X.J. [7]” change to “Chen [7]”.
L148: “Table S2” Should not start with Table S1? Since this was first mentioned. Table S1 is at L209.
L154: “...at 25°C” change to “...and incubated at 25°C”.
L164: “Total flavonoids determination and metabolites analysis” change to “Total flavonoid determination and metabolite analysis”. Same to L298.
L167: “...analysis by HPLC-MS/MS had been described by Su et al [37]” change to “were analyzed using HPLC-MS/MS as described by Su et al. [37]”.
L177: “T.TEST” not “t-test”?
L210: Scientific name should be in italics.
L213: Spacing issue.
L244: “Figure 2C, d-h” Not from “e-h”?
L251-252: Did the authors test different concentrations of flavonoids? Materials and Methods did not mention this, and Figure 3 only shows representatives.
Figure 3: Suggest including a scale bar.
Figure 5B: “...flavonoids content” change to “...flavonoid content”.
L366: Discussion can be further improved. Suggest avoiding repeating what has been discussed in the Result section.
Author Response
Please see the attachment, thank you.
